# A process-based assessment of landscape change and salmon habitat losses in the Chehalis River basin, USA

**Timothy J. Beechie**[1]*, **Caleb Fogel**[2], **Colin Nicol**[2], **Britta Timpane-Padgham**[3]

**1** Fish Ecology Division, National Oceanic and Atmospheric Administration, National Marine Fisheries Service, Northwest Fisheries Science Center, Seattle, Washington, United States of America, **2** Fish Ecology Division, Ocean Associates, Inc., Under Contract to National Oceanic and Atmospheric Administration, National Marine Fisheries Service, Northwest Fisheries Science Center, Seattle, Washington, United States of America, **3** Fish Ecology Division, A.I.S., Inc., Under Contract to National Oceanic and Atmospheric Administration, National Marine Fisheries Service, Northwest Fisheries Science Center, Seattle, Washington, United States of America

\* tim.beechie@noaa.gov

**Data Availability Statement:** The data underlying the results presented in the study will be available on the NOAA Northwest Fisheries Science Center website. https://www.fisheries.noaa.gov/resource/

## Abstract

Identifying necessary stream and watershed restoration actions requires quantifying natural potential habitat conditions to diagnose habitat change and evaluate restoration potential. We used three general methods of quantifying natural potential: historical maps and survey notes, contemporary reference sites, and models. Historical information was available only for the floodplain habitat analysis. We used contemporary reference sites to estimate natural potential habitat conditions for wood abundance, riparian shade, main channel length, and side channel length. For fine sediment, temperature, and beaver ponds we relied on models. We estimated a 90% loss of potential beaver pond area, 91% loss of side-channel length, and 92% loss or degradation of floodplain marshes and ponds. Spawning habitat area change due to wood loss ranged from -23% to -68% across subbasins. Other changes in habitat quantity or quality were smaller—either in magnitude or spatial extent—including rearing habitat areas, stream temperature, and accessible stream length. Historical floodplain habitat mapping provided the highest spatial resolution and certainty in locations and amounts of floodplain habitat lost or degraded, whereas use of the contemporary reference information provided less site specificity for wood abundance and side-channel length change. The models for fine sediment levels and beaver pond areas have the lowest reach-specific certainty, whereas the model of temperature change has higher certainty because it is based on a detailed riparian inventory. Despite uncertainties at the reach level, confidence in subbasin-level estimates of habitat change is moderate to high because accuracy increases as data are aggregated over multiple reaches. Our results show that the largest habitat losses were floodplain and beaver pond habitats, but use of these habitat change results in salmon life-cycle models can illustrate how the potential benefits of alternative habitat restoration actions varies among species with differing habitat preferences.

tool-app/habitat-assessment-and-restoration-planning-harp-model.

**Funding:** Ocean Associates, Inc. provided support in the form of salaries for authors CF and CN. A.I.S., Inc. provided support in the form of salary to BT-P. Northwest Fisheries Science Center provided support in the form of salary to TB. Authors affiliated with Ocean Associates, Inc. and A.I.S., Inc. were under contract to the National Marine Fisheries Service. National Marine Fisheries Service. WDFW provided several data sets. The funders did not have any additional role in the study design, data collection and analysis, decision to publish, or preparation of the manuscript. The specific roles of these authors are articulated in the 'author contributions' section.

**Competing interests:** CF and CN are salaried employees of Ocean Associates, Inc. BT-P is a salaried employee of A.I.S., Inc. There are no patents, products in development or marketed products associated with this research to declare. This does not alter our adherence to PLOS ONE policies on sharing data and materials.

## Introduction

Stream and watershed restoration planning requires some means of identifying and prioritizing restoration actions in order to be cost effective [1, 2]. In a process-based approach, restoration actions focus on restoring natural rates of physical, chemical, and biological process that sustain river ecosystems [3–6]. The main premise of this approach is that degradation of driving processes has caused habitat loss or degradation, and therefore, restoration of watershed processes will restore and sustain habitats and salmon populations over the long term [7–9]. This approach also has the advantages of addressing multiple legal mandates with a single methodology, and avoiding common pitfalls in restoration such as constructing habitats that are not consistent with local physical and biological potential [5]. On the other hand, process-based restoration may have long lag times between implementation and habitat or biological response, and interim habitat construction efforts may also be needed to bridge the gap [5, 10]. In any case, identification of habitat change and root causes of change requires some benchmark against which to compare current conditions, so that we can understand how current condition deviates from potential condition and where restoration potential exists [11, 12].

One long-standing approach to identifying habitat and watershed process changes is to assess habitat change from a reference condition [13–16]. While there is continuing debate over how to identify and quantify such reference conditions [17, 18], there is precedent for assessing change relative to either historical reference conditions [13, 19, 20], or contemporary reference conditions [15, 18, 21]. Both historical and contemporary reference conditions are forms of identifying a "natural potential" for habitat conditions and habitat-forming processes [12, 22]. Importantly, the natural potential against which to assess change is not the same as a restoration target, as it may not be possible to achieve the historical condition [18, 23]. Natural potential is simply a way of describing what is physically and biologically possible, whereas defining restoration targets may include other considerations such as feasibility and desired outcomes that differ from natural potential [12, 18, 24].

In this paper and a companion paper [25], we present a suite of analyses that identify reference conditions, quantify important habitat changes for salmon, and model the potential benefit of restoration actions to salmon populations in the Chehalis River basin. Collectively, we refer to this analysis process as the Habitat Assessment and Restoration Planning model (HARP model). Our assessment focuses on answering two key questions [11, 26]:

1. How have specific habitat features changed from their natural potential conditions?

2. Which habitat changes most constrain rebuilding of salmon populations? (i.e., which restoration actions might provide the greatest benefit to salmon populations?)

The first step in the assessment is analyzing how habitats and habitat-forming processes have changed from natural potential conditions to current conditions. We define natural potential as those conditions that would exist in the absence of human influence, and we assume that natural potential conditions vary spatially as a function of physical and ecological setting. The second step is translating habitat conditions into life-stage capacities and productivities for each species and habitat scenario, where habitat scenarios represent potential habitat improvements from restoration actions. The third step is evaluating the restoration potential of each habitat action using diagnostic habitat scenarios run through salmon life-cycle models. In this paper we describe the first step, and in the companion paper [25] we describe the second and third steps for salmon populations in the Chehalis River basin.

## Methods

Our purpose in this paper is to characterize current and natural potential habitat conditions in the Chehalis River basin to diagnose which habitat restoration actions can most improve abundance and productivity of salmon populations. We focus on eight potential restoration actions, which are the "dials" that we turn in the HARP model to represent management options that can restore processes and improve habitat conditions (Fig 1). Some restoration actions address out-of-channel features and processes (road surface erosion, riparian shade) that influence in-channel habitat attributes, some address in-channel features (wood, bank armor, migration barriers) that influence habitat attributes or access to habitat, and others directly address habitat length or area (channel length, beaver ponds, floodplain connectivity). Each potential restoration action alters one or more habitat conditions, which are quantified as changes in habitat area ($A$) or habitat quality ($\beta$) (Fig 1). Habitat area is the physical area of features such as spawning gravel, pools, or beaver ponds, whereas habitat quality includes habitat attributes such as temperature and fine sediment. Each potential restoration action is then run through the life-cycle models so that we can compare potential restoration outcomes among action types, locations, and species.

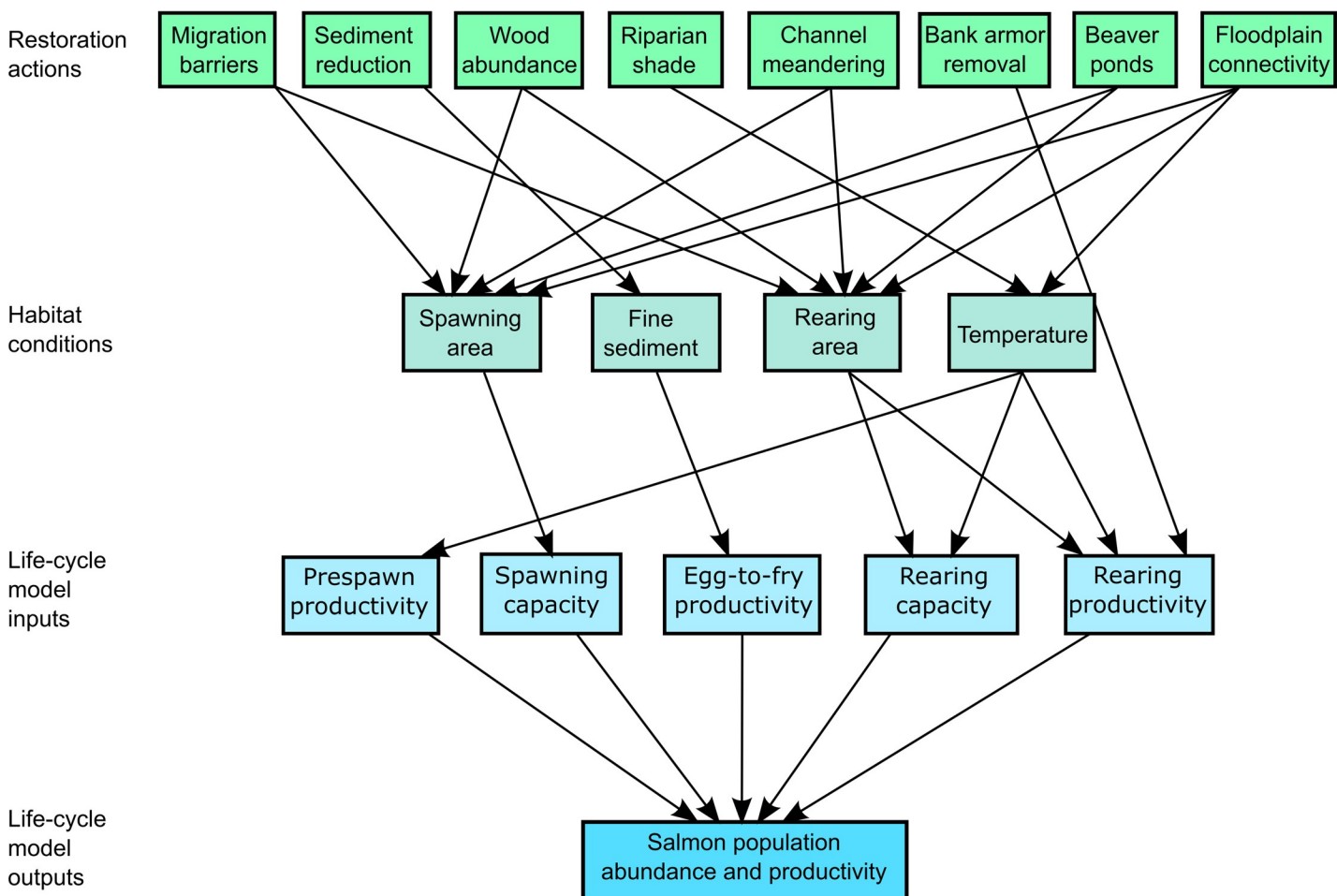

**Fig 1. Conceptual model.** Diagram of model linkages between restoration actions, habitat conditions, life-cycle model input parameters, and salmon population response.

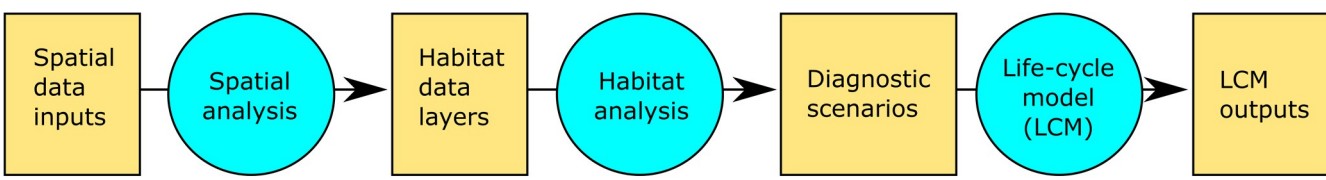

**Raw data layers:**
USGS 10-m DEM
C-CAP land cover raster
PRISM mean annual precip
Washington roads
Ecological Diversity Regions
SWIFD streamlines
WDFW Migration barriers
WDFW Thermalscape
Riparian files
Floodplain habitat polygons
Large river edge habitat
Large river backwaters
Spawning riffles

**Final habitat layers:**
Large river edge habitat
Large river backwaters
Large river spawning riffles
Floodplain habitats
Attributed streamline

**Diagnostic scenarios:**
Migration barriers
Fine sediment
Wood loading
Shade (temperature)
Bank armor
Channel straightening
Beaver ponds
Floodplain habitats

**LCM results**
-Coho
-Spring Chinook
-Fall Chinook
-Steelhead

**Fig 2. Model structure.** The HARP model has three modules (blue circles) that translate geospatial data into habitat data layers, diagnostic habitat scenarios, and life-cycle model outputs (yellow squares).

The HARP model consists of three separate modules: the spatial analysis, the habitat analysis, and the life-cycle models (Fig 2), equivalent to steps one, two and three described above. The spatial analysis includes the GIS processes and analyses that take raw input data (including both publically available and project-specific data layers) and translate those into data layers of current habitat areas and conditions for each reach in the basin. The habitat analysis then takes those data, along with other reference condition information, to create habitat scenarios that include current conditions and historical or potential future conditions that become the inputs to the life cycle models. The habitat scenarios are then run through the salmon life cycle models to evaluate which types of habitat change most influence each species and run-type, and to identify which types of habitat restoration are most likely to aid salmon recovery. Each module can be tailored to local questions and data, making the HARP model a locally adaptable, process-based approach for assessing habitat change and identifying habitat factors that are most important to recovery of Pacific salmon populations.

In the habitat analysis and life-cycle model components of the HARP model [25], the changes in habitat area or quality that we quantify in this paper will alter life-stage capacities ($c$) and productivities ($p$) for each species, which determine density-dependent survival through a life stage using a Beverton-Holt curve [27]. Capacity of each habitat unit ($c_h$) is the product of habitat area and potential fish density ($d$, we use the 95th percentile of observed densities from field studies), scaled by habitat quality:

$$c_h = A \cdot d \cdot \beta$$

Each habitat type has a species- and life-stage-specific density from field studies, and area and quality are derived from habitat data. Life-stage capacity is the sum of capacity across all habitat units. Similar to density, each habitat type has a species and life-stage specific baseline productivity ($p_b$), which is scaled with reach-specific habitat quality to calculate a habitat unit

productivity ($p_h$).

$$p_h = p_b \cdot \beta$$

The habitat quality multipliers range from 0 to 1 (lethal to no effect), as a function of habitat attributes such as temperature or fine sediment. For example, for juvenile coho salmon summer rearing, $\beta = 1$ when the 7-day average daily maximum temperature <17°C, then it declines linearly to $\beta = 0$ at 28°C and higher. Life-stage productivity is the average productivity across all habitat units, weighted by capacity for each unit.

In the last step of the HARP model, these species-specific life-stage capacity and productivity values are integrated into life cycle models to evaluate how changes in habitat area or quality influence salmon populations. Each sub-basin is modeled as a sub-population of a species, with values of $c$ and $p$ for each life-stage aggregated to the sub-basin level (see [25] for details). Each habitat scenario run through the life-cycle models is intended to evaluate how a habitat change from natural potential to current conditions affects restoration potential, or to evaluate how a future restoration and climate change scenario might affect salmon populations. In this paper and the companion paper [25], we focus on influences of past habitat change on restoration potential for salmon populations in the Chehalis River basin, and evaluate how restoration potential various among species and locations. While it is also well known that habitat conditions and salmon populations fluctuate on centennial to inter-annual time scales [28], we do not include temporal variation in habitat conditions in this version of the model so that we can more clearly assess the restoration potential of various actions.

## Study area

The Chehalis River basin drains an area of 6,900 km$^2$ and is underlain by relatively erosion resistant marine basalts and softer sedimentary rocks. The Puget Lobe ice sheet occupied a small portion of the basin ~16,000 years ago near the Black River (Fig 3), and pro-glacial rivers flowing into Chehalis River valley deposited thick glacial outwash deposits in the valleys of the Black River and East Fork Satsop River, creating floodplains that later became extensive marsh and pond habitats [29]. Alpine glaciers from the Cascade Mountains deposited till and outwash deposits in the Newaukum Valley and along the middle Chehalis River, while Alpine glaciers from the Olympic Mountains deposited till and outwash across much of the Humptulips, Wishkah, and Wynoochee River basins [29].

Most of the Chehalis River basin is in commercial or public forest lands (Fig 3). Agricultural lands are concentrated in the low elevation portions of the Black, Skookumchuck, and Newaukum Rivers where glacial melt-water created wide floodplains. Developed lands are somewhat more scattered in the basin, although the majority of developed lands are in the mainstem Chehalis floodplain, Black River basin, and Newaukum River basin.

The Chehalis River has a rainfall-dominated hydrograph with the largest storms and floods generally occurring in fall and winter. Annual rainfall in much of the basin is less than 250 cm/yr [30], but is 250–700 cm/yr in the Olympic Mountains, which exceed 700 m in elevation. The basin is set within the Pacific Coastal Forest region, with dominant tree species including red alder (*Alnus rubra*), black cottonwood (*Populus trichocarpa*), Sitka spruce (*Picea sitchensis*), western hemlock (*Tsuga heterophylla*), western red cedar (*Thuja plicata*), Douglas-fir (*Pseudotsuga menziesii*), and big leaf maple (*Acer macrophyllum*) [31]. Anadromous salmon species in the basin include coho salmon (*Oncorhynchus kisutch*), Chinook salmon (*O. tshawytscha*), steelhead (*O. mykiss*), and chum salmon (*O. keta*). This study focuses on four runs of those species: coho salmon, spring-run Chinook salmon, fall-run Chinook salmon, and steelhead. Spatial distributions of those runs vary, with spring-run Chinook salmon having a very small range, and coho and steelhead occupying the largest ranges (S1 Fig).

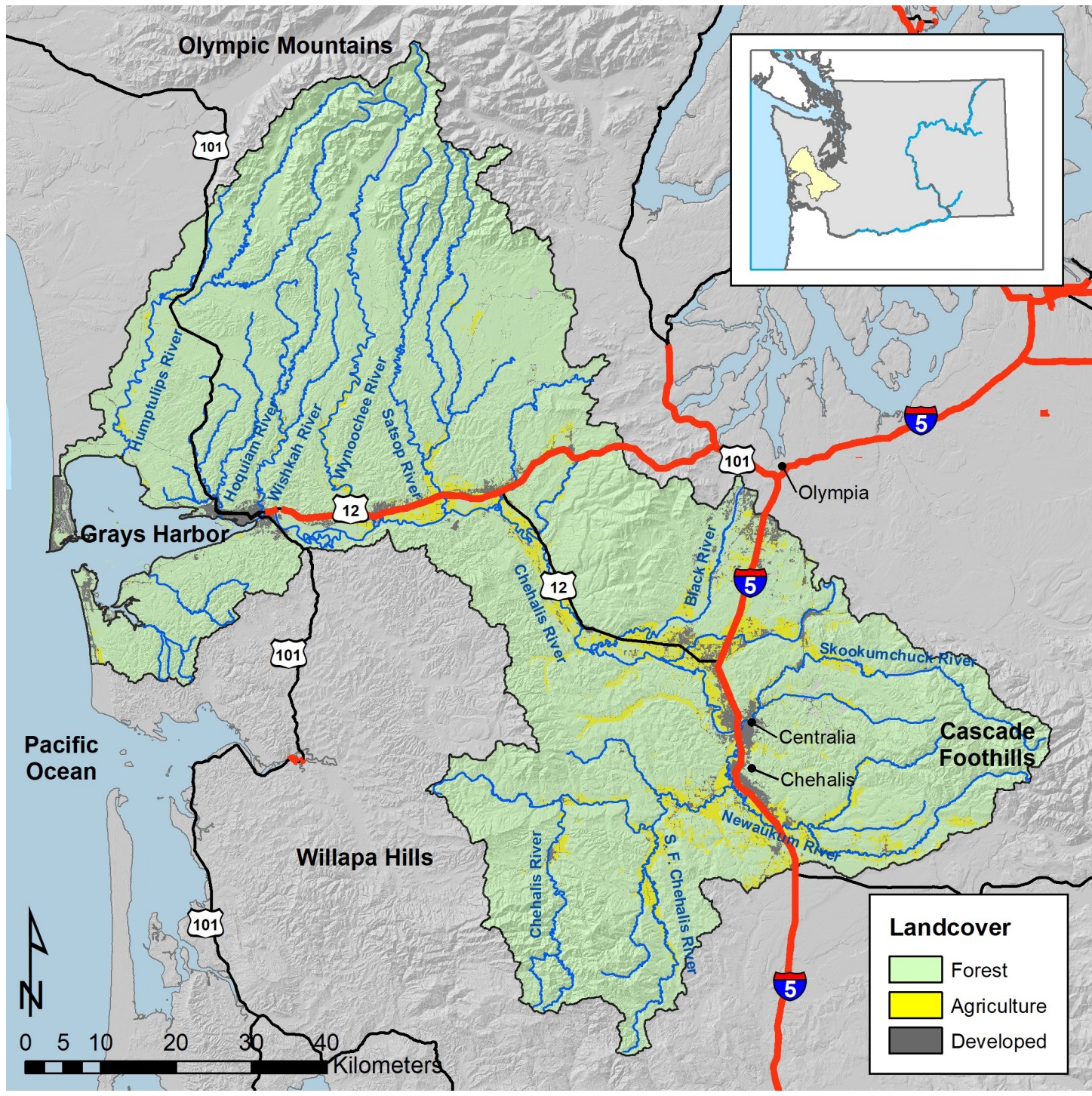

**Fig 3. Study area.** Map of the Chehalis River basin showing major water bodies, tributaries, cities, towns, highways, and mountain ranges.

## Quantifying natural potential condition

A question that is often unstated—and embedded in the idea of setting restoration priorities—is "What do I restore and where?" [1]. Answering this question requires a clear understanding of where habitat conditions are altered from their natural potential, which we evaluate by

quantifying natural potential conditions and then comparing those conditions to current conditions. In this paper we use three general methods for quantifying natural potential condition:

1. historical maps and survey notes,

2. contemporary reference sites, and

3. models.

Historical information has proven valuable for watershed assessment and restoration planning [13, 20]. More specifically, historical information is useful for quantifying reference condition when natural features or attributes change slowly (e.g., over centuries), such as the physical capacity to provide floodplain habitat [32], or for identifying target riparian species [33].

Where historical information is not available (e.g., small stream habitat features or some riparian settings), the second approach of using contemporary, relatively undisturbed sites can be used [13, 21]. In this approach, field or aerial photograph survey information is used to quantify relatively natural conditions across sites stratified by physical or ecological setting [21, 32, 34]. Those conditions can then be extrapolated across a river basin by stratum to estimate natural potential [13].

Finally, where neither historical nor contemporary reference information is available, models can be used to estimate natural potential. One example of this type of analysis is modeling natural potential stream temperatures, for which there are neither historical data nor contemporary reference site data [35]. In this study, we set riparian tree heights to their reference values (based on contemporary references), and then use a shade-temperature model [35] to estimate the natural potential temperature by resetting shade conditions from current condition to natural potential in each reach of the river basin.

## Analyses of habitat change

This study assesses eight potential restoration actions that aim to restore watershed processes and at least partially reverse past habitat degradation. Each restoration action influences salmon migration, spawning, incubation, or rearing through changes in habitat quantity or habitat quality. For example, rebuilding beaver populations or building beaver dam analogs can increase beaver pond area, which is a change in the quantity (area) of a specific habitat type. By contrast, decreasing road density can reduce fine sediment in spawning gravels, which is a change in the quality of spawning gravel (but not the area). For each potential restoration action, we estimate the natural potential habitat condition using one of three methods of quantifying natural potential (Fig 1, Table 1), and then estimate current condition to calculate the potential improvement in either habitat area or quality [13, 36, 37]. Understanding the natural potential condition is essential for understanding whether habitats are degraded or not, and therefore whether restoration is possible. That is, some landscapes or reaches have low natural potential habitat value, and a low current habitat value may simply reflect the natural potential rather than degradation. Landscapes or reaches with high natural potential and poor current condition indicate areas where restoration potential is greatest.

The habitat change analysis uses 14 geospatial data layers as inputs. Five of the geospatial inputs are publically available data sets, including topography, land cover, land use, precipitation, and road locations. Four of the geospatial data sets were produced or updated specifically for use in habitat restoration planning for the Chehalis River basin: spatial units, stream lines, migration barriers, and current stream temperatures. Spatial units defined for the salmon and steelhead models include 63 subbasins, which were then grouped into 10 Ecological Regions (Fig 4). The 63 subbasins are either independent tributaries entering the mainstem Chehalis

**Table 1. Reference condition methods.**

| Assessment component | Analysis type | Method for defining natural potential |
|---|---|---|
| **Floodplain connectivity** | Historical information (marshes, ponds) | Mapped based on General Land Office surveys of 1853 to 1901 |
| | Contemporary reference (main-channel and side-channel length) | Reach-specific natural potential side-channel length estimated from contemporary reference sites [34] |
| | Model (temperature) | Temperature effect based on model of floodplain connection effect on temperature [38] |
| **Main channel length** | Contemporary reference | Reach-specific natural potential main-channel length estimated from contemporary reference sites [34] |
| **Main channel bank armor** | NA (bank armor assumed not present historically) | Bank armor removed from stream network for analysis |
| **Wood abundance** | Contemporary reference | Reference spawning and rearing habitat areas extrapolated from contemporary reference sites; data stratified by channel slope (small stream) [13] |
| | Contemporary reference | Natural potential wood cover extrapolated from Queets River observations (large river) |
| **Riparian shade** | Contemporary reference (shade) | Natural potential shade based on reference tree heights for non-floodplain and floodplain channels |
| | Model (temperature) | Modeled natural potential temperature using a shade-temperature model [35] |
| **Road density (fine sediment)** | Model | Fine sediment levels modeled based on unpaved road density (road density set to zero for natural potential) |
| **Beaver pond area** | Model | Natural potential based on modeled beaver dam frequency and size [39] |
| **Migration barriers** | NA (artificial barriers assumed not present historically) | Artificial migration barriers removed from stream network for analysis |

Summary of methods for defining natural potential condition for each assessment component.

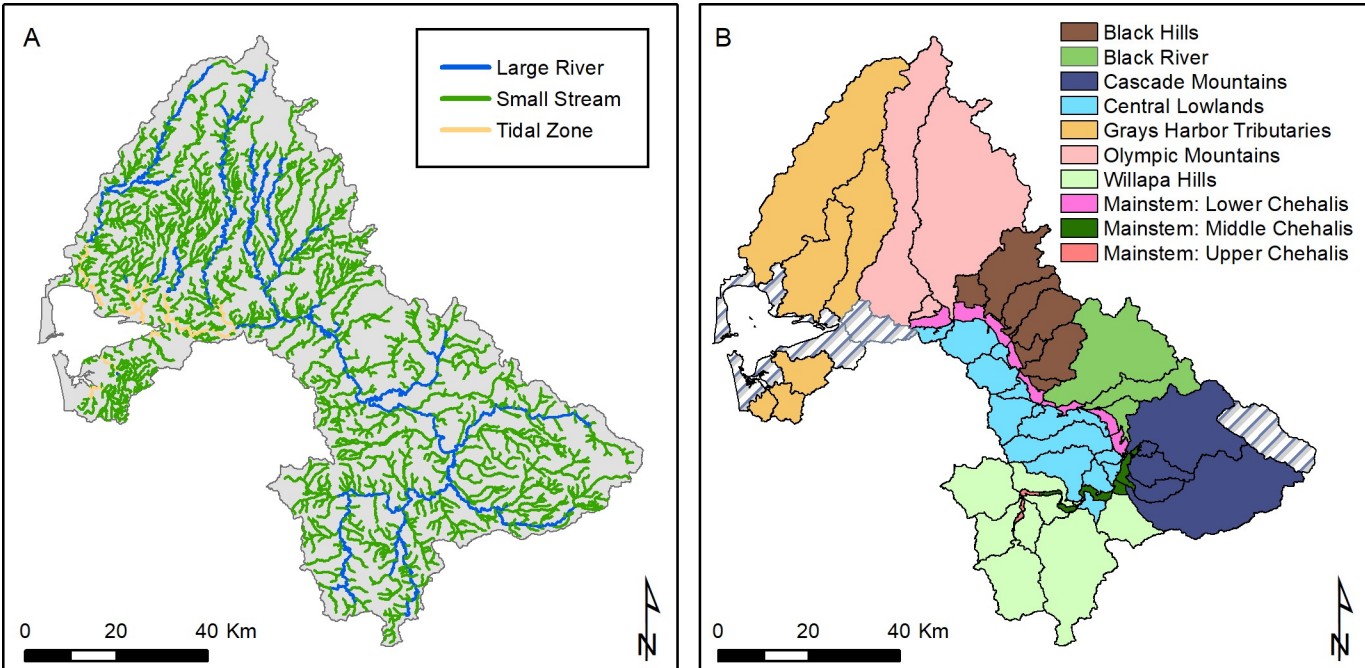

**Fig 4. Model spatial structure.** Distribution of large rivers (bankfull width >20 m) and small streams (<20 m bankfull width) in the Chehalis River basin (left panel). Map of the 63 subbasins (black boundaries) and Ecological Regions (colored regions) (right panel). Gray-striped subbasins are not included in Ecological Regions.

(which vary in size from small streams to large tributary rivers), or sub-reaches of the mainstem Chehalis River. We also assess habitat change at the scale of smaller Geospatial Units (GSUs) within subbasins (not shown in Fig 4) so that restoration potentials can be compared across units of more similar size.

The stream lines were modified slightly from the National Hydrography Dataset (NHD) (https://www.usgs.gov/core-science-systems/ngp/national-hydrography/) for accuracy, and were attributed with summer and winter wetted widths and spawning distributions for each salmon and steelhead species (ICF International, unpublished data). For our analysis we segmented this layer into 200-m reaches. We used Washington Department of Fish and Wildlife (WDFW) data for migration barriers (https://geodataservices.wdfw.wa.gov/hp/fishpassage/index.html) and summer stream temperature [40]. We produced the remaining five data sets as part of the habitat change analysis, including digitized habitat features for current large river habitat (3 data sets), historical floodplain habitat, and current riparian condition. Each data set is described in the following methods.

**Habitat types.** We selected a habitat typing system in which each habitat type is 1) sensitive to change due to land use or restoration actions, and 2) a good predictor of salmonid fish density or survival [11]. Most juvenile salmonids occupy relatively shallow and low-velocity areas for rearing, and exhibit specific cover-type preferences [41, 42]. Adult salmonids also demonstrate consistent spawning habitat depth and velocity preferences, which vary by species [43–45]. However, locations of those rearing and spawning habitats vary among stream sizes, and habitat types and inventory methods differ between small streams, large rivers, and off-channel habitats [41, 42]. Therefore, we used separate habitat typing systems for small streams (<20 m bankfull width), large rivers (>20 m bankfull width), and floodplains (Table 2).

Table 2. Habitat typing system.

| Macro habitat type | Habitat type | Definition |
|---|---|---|
| Small stream | Riffle | Shallow, fast water (typically >0.45 m/sec) |
| | Pool | Deep, slow water (typically <0.45 m/sec) |
| | Beaver pond | Beaver pond with median size 500 m$^2$ |
| Large river | Bank edge | Vertical or steeply sloping shore, velocity <0.45 m/sec, depth <1.0 m, no bank armor |
| | Armored bank edge | Vertical or steeply sloping shore, velocity <0.45 m/sec, depth <1.0 m, banks are armored |
| | Bar edge | Gently sloping shore, velocity <0.45 m/sec, depth <1.0 m |
| | Backwater | Partially enclosed areas separated from the main river channel, velocity <0.45 m/sec |
| | Mid-channel | All habitat area not included in bank and backwater habitats, often >1 m deep or velocity >0.45 m/sec |
| Floodplain | Marsh | Partially vegetated, dry in summer and wet in winter |
| | Pond (small) | Open water, wet year-round, <500 m$^2$ |
| | Pond (large) | Open water, wet year-round, 500 m$^2$ to 5 ha |
| | Lake | Open water, wet year-round, >5 ha |
| | Slough | Side channel with pond-like habitat |
| | Side-channel riffle | Shallow, fast water (typically >0.45 m/sec) |
| | Side-channel pool | Deep, slow water (typically <0.45 m/sec) |

Definitions of habitat types used to estimate rearing habitat capacity and productivity. Spawning gravel area is typed separately.

To distinguish large rivers from small streams, we segmented the stream lines into 200-m reaches, then used modeled bankfull width to distinguish channels <20 m and >20 m modeled width. Bankfull width was modeled based on drainage area ($A$) and mean annual precipitation ($P$) upstream of each segment, using an equation form similar to that used for estimating bankfull discharge [46]. Because our observations showed that rivers in alpine areas were much wider at a given drainage area than those in non-alpine areas, we developed separate empirical prediction equations using measured bankfull widths for subbasins with headwaters in alpine areas (Wynoochee and Humptulips subbasins, n = 31), and for all other subbasins (n = 59):

$$Alpine\ areas:\ \text{Bankfull width} = 0.000015(A)^{0.377}(P)^{2.164},\ r^2 = 0.59$$

$$All\ other\ areas:\ \text{Bankfull width} = 0.11(A)^{0.347}(P)^{0.745},\ r^2 = 0.74$$

Drainage area upstream of each reach was calculated using flow accumulation with the 10-m National Elevation Dataset (NED, https://ned.usgs.gov), and mean annual precipitation upstream of each reach was calculated using a weighted flow accumulation of the mean annual precipitation grid [30]. Channels near the threshold width of 20 m were sometimes classified incorrectly, so the classification was manually corrected where air-photo-measured channel width was substantially different from the predicted width. Floodplain habitats include all remaining habitat types outside of small streams and large rivers: marshes, ponds, lakes, side-channels, and sloughs.

**Floodplain habitat.** We used historical information and contemporary reference site data to calculate natural potential floodplain habitat areas in the Chehalis River Basin. The primary data sources were the General Land Office (GLO) cadastral survey maps and notes dating from 1853 to 1901 (https://www.blm.gov/or/landrecords/survey/ySrvy1.php), and the NHD 1:24,000 scale waterbodies (https://www.usgs.gov/core-science-systems/ngp/national-hydrography/). We first digitized historical features from the GLO survey maps, then used the GLO survey notes to refine polygon boundaries and classify them by type (e.g., marsh, pond, lake). Many features that exist today were missing from those maps and survey notes, mainly because those features were located in un-surveyed areas between surveyed section lines. Therefore, we also added and classified NHD waterbody features as historical habitats if they appeared to be in a relatively natural state, under the assumption that contemporary features (except artificial features like retention ponds), likely existed historically in some form. Each waterbody polygon was classified as being present during historical, current, or both historical and current periods. Current habitat condition was evaluated using aerial imagery, lidar, and geo-referenced fish barrier data (https://geodataservices.wdfw.wa.gov/hp/fishpassage/index.html), and habitat condition was classified as natural, inaccessible due to a migration barrier, modified riparian vegetation, or physically modified.

The most common floodplain habitat types were marshes, ponds and side channels. GLO survey notes indicated that marshes were typically dry in late summer and early fall when the surveys were conducted, and also noted that those areas had standing water in winter. Therefore, marsh areas were assigned a habitat area of 0 in summer and the area of the polygon in winter. Marshes were mostly vegetated, indicating that even in winter the actual habitat area is smaller than the polygon area, which is accounted for in the rearing densities for these habitats [25]. Ponds were wet at the time of the survey, and were considered to have the same wetted area as the polygon area in both summer and winter. Side channels were treated as pool-riffle habitats similar to small stream channels (described later). The survey notes sometimes used

the term slough to describe a side-channel or oxbow, which we considered to be pond-like habitat [13].

We considered lost side-channel habitats due to channelization and levees to be part of the floodplain habitat change, but those losses could not be quantified from the GLO surveys. Therefore, we relied on side-channel length data for each channel pattern from contemporary reference sites [34], and used those data to create reach-specific natural potential side channel lengths for each mainstem reach based on its channel pattern (S1 Table).

**Wood abundance.**   In our model, wood influences areas of habitat units such as pools as well as area of spawning gravel (habitat quantity). Wood also modifies habitat specific density and productivity (habitat quality) [25]. We relied on field data to characterize current habitat conditions in small streams that are covered by tree canopy and therefore cannot be surveyed with aerial imagery. For current habitat areas, we assumed that measured habitat conditions reflected current (low) wood abundance, and that habitat conditions in reference sites reflected high wood abundance. For current pool areas, we used 339 recent habitat surveys conducted by WDFW in small streams of the Chehalis River basin from 1999 to 2014, classified by mean reach slope and adjacent land cover class (from NOAA's C-CAP 30-m resolution land cover data 2016) (https://coast.noaa.gov/digitalcoast/data/home.html). We calculated mean percent pool area for each slope and land cover class to create a matrix of mean percent pool for current conditions in small streams (S2 Table). We then extrapolated the average current percent pool values for each slope-cover class to all similarly classified small stream segments [13, 47, 48]. For natural potential pool areas, we extrapolated mean percent pool values from contemporary reference site data to small streams in the stream layer by slope class [13, 47] (S2 Table). Finally, for both natural potential and current conditions, we translated percent pool in each reach into pool area by multiplying percent pool by reach- and season-specific wetted widths (ICF International, unpublished data) and reach length. All remaining area in each reach was classified as riffle habitat. Where reaches overlapped floodplain pond or lake habitat polygons, we set both pool and riffle area to 0 to avoid double counting the area in both small stream and lake or pond habitat.

We calculated the total spawning area in each small-stream reach using:

$$Spawning\ Area = \#pools \times wetted\ width \times tail\ crest\ length$$

which assumes that spawning occurs on riffles at pool tail crests and not the entire length of long riffles. We set the tail crest length at ½ the wetted width based on our aerial photograph observations in large rivers in the Chehalis basin. The number of pools in each reach is calculated as:

$$\#pools = reach\ length/(pool\ spacing \times wetted\ width)$$

where pool spacing is in units of wetted widths/pool, and is a function of channel slope and wood abundance. We converted the original data of pool spacing in bankfull widths per pool [49, 50] to pool spacing in wetted widths per pool, by dividing BFW/pool by 0.4, which is the ratio of wetted width to bankfull width from our Chehalis-specific width prediction equations. For current conditions (low wood abundance), pool spacing for low-slope and high-slope channels was:

- slope < 1% and low wood: 12.5 wetted widths/pool

- slope > 1% and low wood: 27.5 wetted widths/pool

For natural potential conditions (high wood abundance), pool spacing for low-slope and high-slope channels was:

- slope < 1% and high wood: 6.25 wetted widths/pool

- slope > 1% and high wood: 5 wetted widths/pool

In large rivers (bankfull width >20 m), we digitized bar edge habitats (sand, gravel, or boulder) and natural and armored bank edge habitats as lines, and backwater pools as polygons, using high resolution (30 cm) Google imagery in ArcMap GIS at 1:3,000 scale or closer. We then converted bank and bar edge habitat lengths to habitat areas using a field-surveyed relationship between total wetted width and edge habitat width (S2 Fig). Current condition habitat areas in each reach were the sum of all current bank, bar, and backwater habitat areas (including armored bank habitats). All remaining wetted area was classified as mid-channel habitat. Current wood loading was assumed to be low in all habitat types based on local observations, and we assumed that observed fish densities reflected currently low wood abundances [25]. For natural potential, we used the nearby Queets River (within Olympic National Park) as a contemporary reference site for natural wood loading. Based on aerial imagery, we estimated that 5% of edge habitats had wood cover for the natural potential condition. Higher natural wood loading in large rivers increases juvenile salmon density [42], so in the model we increase fish density and rearing productivity for all species, but wood does not affect rearing habitat area in large rivers.

For large-river spawning habitat, we digitized riffles from recent aerial imagery for all large river reaches to represent current conditions. We assumed that spawning occurred on pool tail crests and not the entire length of riffles, and we used the same equation to calculate spawning area as we did for small streams but where the number of pool tails was manually digitized. For natural potential conditions, we found no data to support a specific percent increase in large river spawning gravel area as a function of wood abundance. However, we know that in small streams with low slope and high wood abundance, bar frequency is double that of small streams with low wood abundance (described above). We also know that wood in large channels is mostly mobile and accumulates as bar apex or meander bend jams, which have less effect on pool and bar frequency than wood in small channels [10, 51]. Therefore, we assumed that increased spawning area in large rivers is much less than the doubling in small streams, and we modeled a spawning area increase of 30% at high wood abundance in large rivers, which is intended to reflect increased spawning gravel retention and holding pool formation.

**Main channel length and bank condition.** To estimate changes in channel length and bank condition, we calculated sinuosity of each main channel reach and compared it to the sinuosity of contemporary reference reaches with a similar channel pattern [34] to calculate reach-specific ratios of historical length to current length (S1 Table). This ratio is used as a main channel length multiplier to increase habitat unit areas by a fixed percentage for each reach, assuming that habitat unit widths did not change due to channel straightening. All armored large river bank habitats that were digitized from aerial imagery were also reset to natural bank habitats to estimate natural potential, increasing habitat quality (fish densities) for those habitats.

**Riparian shade.** The riparian analysis evaluated changes in tree height and canopy opening angle (a surrogate for shade) by comparing current riparian conditions to natural potential conditions based on contemporary reference site data [35]. We assessed current tree heights and buffer width using lidar tree height data where available [35], and using aerial photography where lidar data were unavailable. For aerial photography, we binned trees into three size classes based on crown diameter, calculated tree heights from lidar at over 400 calibration points distributed across the size classes, and used the median tree height for each size class as the estimate of current tree height for all aerial photograph sample points (S3 Fig). We estimated natural potential tree heights using separate contemporary reference conditions for

small streams which, we equated to non-floodplain channels (52 m tree height), and large rivers, which we equated to floodplain channels (30.5 m tree height). Detailed descriptions of species and age distributions in contemporary riparian reference sites are in S4 and S5 Figs.

Current canopy opening angle was calculated using the canopy opening width and current tree heights on each side of the stream (S6 Fig). Analysis points are at 10-meter spacing along the stream where we used lidar, and a maximum 300-meter spacing where we used aerial imagery. Natural potential canopy opening angle was calculated at each point using the reference tree heights and the current canopy opening width for all small stream and large river segments.

For current temperature conditions, we use the WDFW Chehalis Thermalscape model estimates of August average daily average (August ADA) temperature for each 1-km long reach in the basin [40]. To assess the influence of temperature on habitat quality, we first converted the August ADA to more relevant temperature metrics. We used the August seven-day average daily maximum (7-DADM) for coho and steelhead summer rearing and spring-run Chinook pre-spawning, and the June 1–21 average daily maximum (June 1–21 ADM) for spring- and fall-run Chinook outmigration. We used the measured stream temperatures that informed the Thermalscape model to calculate the three temperature metrics (August ADA, 7-DADM, and June 1–21 ADM) at 101 unique sites over 3 years (28 sites in 2014, in 70 sites in 2015, and 45 sites in 2016). We then regressed the 7-DADM and the June 1–21 ADM against the August ADA which allowed us to convert the reach-specific Thermalscape temperatures (August ADAs) to reach-specific estimates of the other temperature metrics:

$$7-\mathrm{DADM} = 1.18 \cdot \mathrm{Aug\ ADA} + 1.01,\ \mathrm{and}$$

$$\mathrm{Jun}1-21\ \mathrm{ADM} = 1.12 \cdot \mathrm{Aug\ ADA} - 2.23.$$

Finally, we used an empirical temperature model to estimate the change in temperature from natural potential due to loss of canopy cover:

$$\Delta 7-\mathrm{DADM} = 0.035 \cdot \Delta\theta$$

where Δ7-DADM is the change in 7-DADM and Δθ is the change in canopy opening angle. Because the above equation is used to relate canopy cover to 7-DADM, we empirically related change in the Jun1-21ADM to a change in 7-DADM to estimate change in the Jun1-21ADM:

$$\Delta\mathrm{Jun}1-21\ \mathrm{ADM} = 0.98\ (\Delta 7-\mathrm{DADM}).$$

**Fine sediment.**   An estimate of the current fine sediment (<0.85 mm) condition in each reach was calculated from a shear stress index and road density. The shear stress index is an indicator of stream energy to transport sediment, and is calculated as channel slope multiplied by bankfull width [52]. We calculated the shear stress index of each reach in which we had fine sediment data, and plotted local fine sediment measurements [53] against that index. We found that reaches with shear stress index ≤0.05 had consistently high fine sediment percentages regardless of road density, and we assigned those reaches the average percent fine sediment of low-energy reaches (27%) (S7 Fig).

In reaches with shear stress index >0.05, we assumed that percent fines could be modeled as a function of road density based on a nearby study from the Queets River basin [54]. We used a Washington Department of Natural Resources roads layer (https://fortress.wa.gov/dnr/adminsa/gisdata/metadata/road.html) and the Washington state land use layer (http://www.ecy.wa.gov/services/gis/data/planningCadastre/landuse.htm) to create a layer of unpaved

roads in forest lands. Road density was then used to estimate percent fine sediment in spawning gravels for each reach using a regression relationship between the fraction of fine sediment in a stream segment and the road density within the drainage area above that segment (S7 Fig) [54]:

$$fine\ sediment = 5.74 + 2.05(percent\ road\ area),\ (\text{r}^2 = 0.62)$$

Where *fine sediment* is the percent fine sediment <0.85 mm, and *percent road area* is the percent of the basin area covered by unpaved road surfaces. This approach does not incorporate the influence of other sediment sources on fine sediment levels, so fine sediment levels in many locations will likely differ from modeled fine sediment levels based on road density alone.

For natural potential conditions, we assumed that the low-energy reaches (shear stress index ≤0.05) had naturally high levels of fine sediment, and we assigned them the same value as the current condition (27% fines) (S7 Fig). For reaches with shear stress index >0.05, we set the road density to zero, so reaches were assigned a modeled value of 5.7% fine sediment (the intercept of the regression equation in the fine sediment model).

**Beaver pond habitat.** We estimated change in beaver pond area for small streams (<20 m bankfull width) only, assuming that beaver dams do not persist in large rivers [39], and that the floodplain habitat data would include ponds and marshes on floodplains. In small streams, current beaver dam densities ranged from 0.1 to 1.1 ponds/km among subbasins, with a length-weighted average of 0.6 ponds/km [55] (S3 Table). We reduced the density to 0.55 to avoid double counting of modeled beaver ponds with ponds already in the floodplain habitat data set. To account for inundation of pools and riffles by ponds, we used a typical pond length of 25 m, which inundates 1.5% of the small stream length and reduces pool and riffle areas by that percentage in the current condition.

We estimated natural potential pond area in small streams using a pond frequency of 6 ponds/km and a median pond area of 500 m$^2$ [39], which is equivalent to 3,000 m$^2$ of pond area per km of stream. The assumption of 6 ponds/km is conservative but still 10× the current pond frequency, and it is lower than most frequencies observed where there are relatively undisturbed beaver populations [39]. To account for inundation of pools and riffles by ponds in the natural potential condition, we used the same pond length of 25 m and reduced pool and riffle areas by 15% for the natural potential condition.

**Migration barriers.** Migration barrier data were from a WDFW database, which was recently updated for this analysis (https://geodataservices.wdfw.wa.gov/hp/fishpassage/index.html). We used this data set for estimating reductions in spawning and rearing capacities for salmon and steelhead due to restricted access by barriers. The database contained 1790 barriers, classified as culverts, dams, waterfalls, or not specified. Approximately 90% of barriers have a fish passage rating assigned by WDFW (usually 0, 0.33, 0.67, or 1.0). For the remaining barriers, the database assumes a fish passage rating of 0.5 based on the recommendation of WDFW.

In the models, the barrier passage rating reduces the estimated capacity or productivity of each upstream reach [25], and the cumulative passage rating for each reach is the product of the passage ratings of all downstream barriers.

$$\beta_{passage} = \prod \beta_b,$$

where $\beta_{passage}$ is the cumulative passage multiplier for each reach and $\beta_b$ is the passage rating of each barrier downstream of a given reach.

For the natural potential condition, we set all passage ratings for man-made barriers to one. This eliminates all artificial migration barriers from the analysis, and sets all habitat capacities and productivities to their potential value in the absence of barriers.

## Results

In decreasing order of magnitude, habitat changes included loss of side-channels, beaver ponds, and floodplain marshes; decreased rearing pool and spawning gravel area due to reduced wood abundance; reduced shade; blocked habitat by migration barriers; decreased spawning gravel quality due to increased fine sediment; and large river bank armor and channel straightening (Table 3). Notably, all of the slow-water and off-channel habitats (beaver ponds, marshes, side-channels) have declined by roughly 90%. Only 8% of estimated historical floodplain marshes and ponds remain in relatively natural condition today, and roughly half of the historical floodplain habitat area has been completely lost from the landscape (Fig 5). The remaining 41% exists in a degraded condition. Floodplain disconnection has also eliminated an estimated 241 km of the natural potential 266 km of side-channel length (-91%) (Table 4).

We estimated that loss of instream wood has decreased spawning gravel area by 23 to 68% across subbasins (Table 3). For rearing habitats, the estimated change in pool area in small streams due to wood loss is smaller because low slope channels have relatively high current pool areas and the Chehalis basin has extensive lengths of low-slope small streams. Rearing habitat area in large rivers is unchanged in our analysis (i.e., decreasing wood abundance does not affect pool or edge habitat areas due to lack of data on the relationship), but rearing habitat quality has been reduced due to removal of wood cover from edge habitat units.

**Table 3. Summary of habitat changes in the Chehalis River basin and Grays Harbor tributaries.**

| Habitat type or attribute | Change |
|---|---|
| **Floodplain habitat** | • Side channel length decreased by 91% |
| | • Marshes and ponds: 51% lost, 41% degraded, 8% intact |
| **Beaver ponds** | Decreased by 90% in small streams (<20m bankfull width) |
| **Wood: spawning habitat area** | Decreased 23% to 68% among subbasins |
| **Shade: temperature increase** | Length of channel with estimated temperature increase >2˚C from historical to current: 16% |
| **Migration barriers** | Percent of spawning habitat above barriers (mostly partial) |
| | • Spring-run Chinook: 0% |
| | • Fall-run Chinook: 10% |
| | • Coho: 22% |
| | • Steelhead: 18% |
| **Fine sediment** | Mean modeled percent fines (basin-wide) |
| | • Historical: 13.7% |
| | • Current: 18.1% |
| **Large river channel length** | Decreased by 5% (>20m bankfull width only) |
| **Bank armoring** | 7% of large river banks armored (>20m bankfull width only) |

All metrics refer to the combined extent of all salmon and steelhead spawning and rearing habitat, unless species are listed separately. Note that the same metric of change cannot be used for all habitat types or attributes. For example, loss of floodplain habitat area and side-channel length are simply the percent decrease from historical levels, whereas temperature is expressed in terms of percent of channel length above a specified threshold of temperature change.

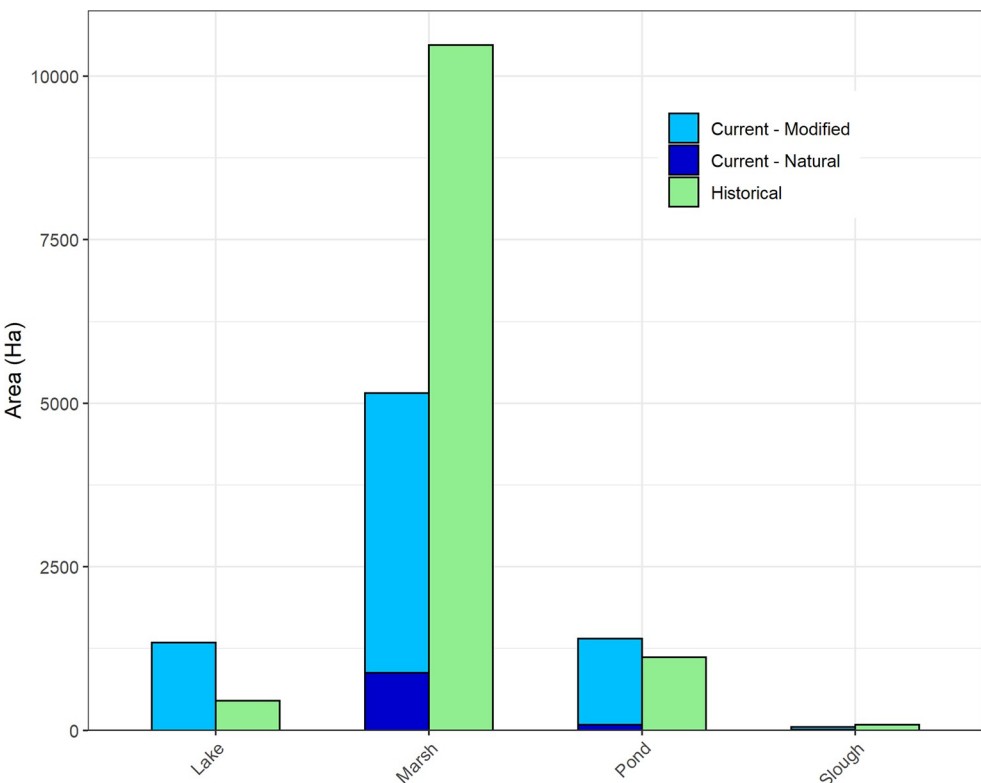

**Fig 5. Floodplain habitat change.** Area of floodplain habitat lost and gained (current floodplain area–natural potential floodplain area). Increases in lakes and ponds are due to constructed reservoirs, retention ponds, and other small impoundments.

A significant portion of the stream length has a current canopy opening angle that is wider than the natural potential condition (Fig 6). In particular, reaches with canopy opening angle >100˚ have increased from 3% of total reach length to 18% of total length. As a result, summer stream temperatures have increased substantially in some areas, but only 16% of total stream length has a modeled temperature increase of more than 2˚C (Table 3).

Table 4. Changes in channel length.

| Ecological Region | Loss of main channel length | Loss of side channel length |
|---|---|---|
| **Black Hills** | -0 km (0%) | -0 km (0%) |
| **Black River** | -0.2 km (0.5%) | -0.8 km (79%) |
| **Cascade Mountains** | -11.1 km (10.7%) | -44.7 km (97%) |
| **Central Lowlands** | -0.02 km (1%) | -0.1 km (100%) |
| **Grays Harbor Tribs** | -7.6 km (5.2%) | -58.2 km (88%) |
| **Mainstem: Lower** | -5.8 km (6.1%) | -19.4 km (84%) |
| **Mainstem: Middle** | -3.6 km (6.4%) | -16.4 km (99%) |
| **Mainstem: Upper** | -0 km (0%) | -0 km (0%) |
| **Olympic Mountains** | -14.1 km (5.2%) | -101.7 km (93%) |
| **Willapa Hills** | -0 km (0%) | -0 km (0%) |
| **Total** | -42.4 km (4.9%) | -241.3 km (91%) |

Loss of main channel length and side-channel length, by Ecological Region.

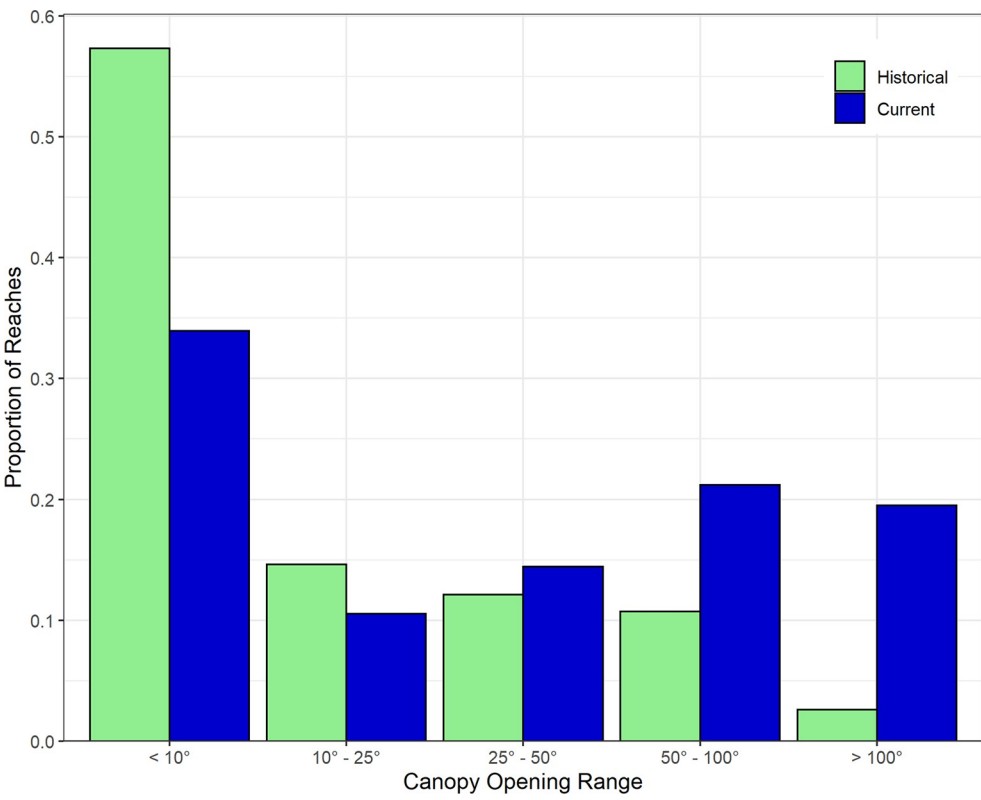

**Fig 6. Change in stream shade (canopy opening angle).** Frequency distributions (proportion of total reaches) of natural potential and current canopy opening angles in the Chehalis River basin. Note the large increase in very wide canopy opening angles (>100˚) and large decrease in narrow angles (<10˚).

Effects of migration barriers on access to spawning habitat varied significantly by species. For example, for spring-run Chinook—a species with a spawning range limited to large rivers —barriers do not block any spawning habitat within the range documented in the fish distribution data layer. By contrast, 22% of coho spawning habitat and 18% of steelhead spawning habitat length is above artificial barriers. However, most barriers are not 100% blocking, so effective spawning habitat loss is smaller than the total length of habitat above barriers.

Where there has been channel straightening, reference site data applied on a reach-by-reach basis show that 42 km of the estimated 860 km of natural potential large river main channels has been lost (-5%) (Table 3). Our mainstem habitat mapping showed that 7% of large river banks are armored today.

Distributions of restoration potentials vary spatially, and the patterns of variation differ depending on habitat attribute (Fig 7). For example, fine sediment changes are most pronounced in headwater basins where land use is predominantly commercial forest, whereas temperature changes are most pronounced in lower subbasins with predominantly agricultural or urban development. Some patterns also reflect the underlying physical potential of the landscape, such as floodplain habitat loss. Most of the historical floodplain habitat was in a few subbasins heavily influenced by glacial outflows during the last glacial period (S8 Fig), and the distribution of habitat losses reflects both high natural potential and significant habitat loss due to land uses that drained historical floodplain habitats.

The beaver pond model assumed an empirically-estimated fixed density of beaver ponds for current condition, and a fixed density for historical conditions. Therefore, the estimated

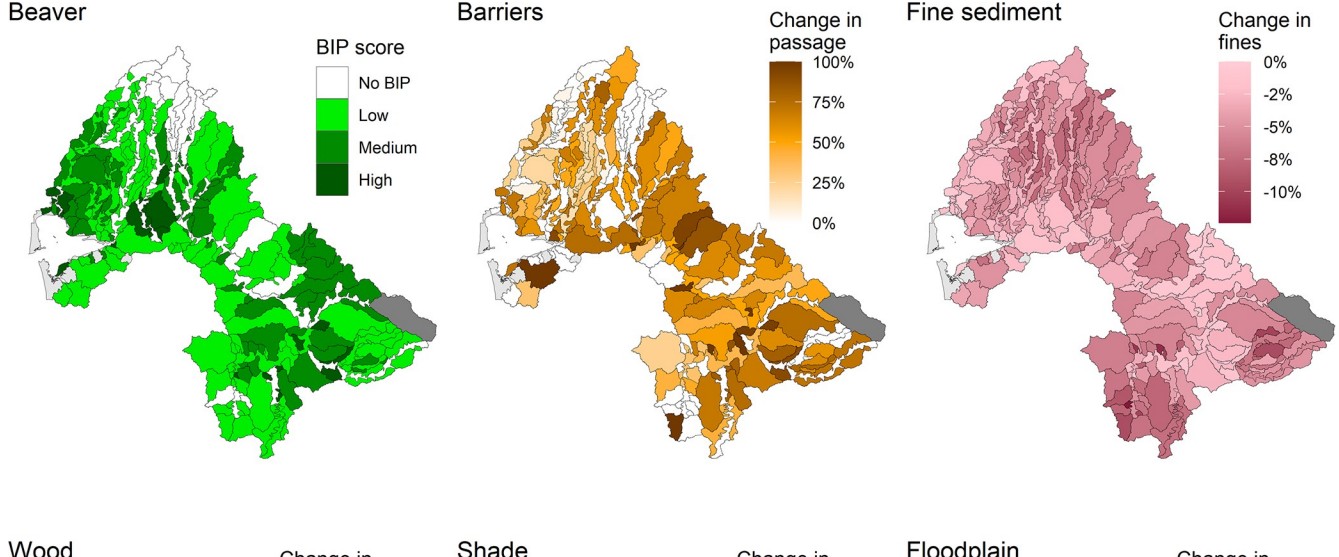

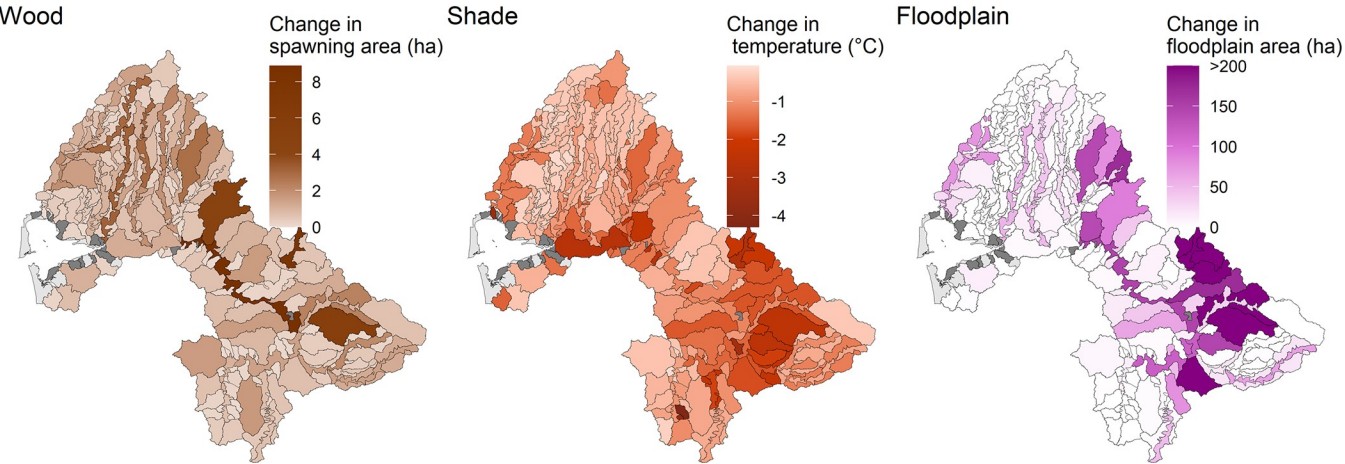

**Fig 7. Spatial distribution of restoration potentials.** Maps of restoration potential for six habitat attributes under current climate conditions and development, indicating spatial variation in restoration potential without effects of climate change or future development. **Beaver** is the mean beaver intrinsic potential score by Geospatial Unit (GSU). **Barriers** is the potential increase in mean passability percentage by GSU if migration barriers are removed. **Fine sediment** is the potential decrease in fine sediment if forest road surface erosion is reduced (mean historical percent fines–mean current percent fines). **Wood** is the potential increase in hectares of spawning gravel by GSU via increasing wood abundance (historical spawning gravel area–current spawning gravel area, in hectares). **Shade** shows the stream temperature restoration potential (Δ˚C) by GSU via riparian shade restoration (mean historical ˚C–mean current ˚C). **Floodplain** indicates the potential increase in hectares of floodplain habitat area by GSU via restoration of floodplain connectivity (historical ha–current ha of floodplain habitat).

percent decrease in beaver pond area was the same across all reaches and subbasins. Based on this simple model, we estimated that the number of beaver ponds in small streams has decreased by 90%, and beaver pond area has decreased by more than 950 ha from its natural potential area in the basin.

## Discussion

In this study we aimed to advance the practice of habitat change assessment by addressing a wide array of habitat types or attributes in the HARP model, using a combination of historical data, contemporary reference sites, and models. Other studies have focused on one habitat type or attribute using one method [20, 33], or evaluated several attributes using a single approach [36, 37, 56, 57]. To our knowledge, no single assessment has evaluated as many

causes of habitat change using all three general approaches to quantifying restoration potential. By using all three general approaches in the spatial analysis component of the HARP model, we were able to develop a comprehensive and quantitative picture of how riverine and floodplain habitats have been altered from their natural potentials in the Chehalis River Basin. However, confidence in the results varies due to uncertainties inherent in the different approaches. Nonetheless, the large differences in estimated habitat losses suggest that we should be confident in the general ranking of magnitudes of habitat loss.

Other studies reconstructing historical conditions for one or more attributes in the region showed similar habitat changes. For example, more than 80% of floodplain and estuarine habitat losses are common in the region [13, 58], whereas severely degraded riparian areas are often less than 20% of riparian length [59, 60]. However, some attributes, such as migration barriers, vary widely in their influence on habitat conditions depending on location. For example, in the Willamette and Elwha River basins, dams block 40 to 90% of salmon habitat [61, 62], whereas culverts and other small migration barriers typically block a small percentage of salmon habitats in other river basins [13, 62]. The Chehalis basin has two relatively large dams, but they are high in tributary watersheds and block a relatively small amount of habitat compared to culverts and other small barriers.

## Quantifying natural potential

We were only able to use historical information to characterize natural potential condition for large floodplain habitat features that intersected GLO survey lines, but the historical approach provided relatively high confidence in site-specific estimates of habitat change. Floodplain marshes and ponds were mapped and recorded in the GLO notes, revealing hundreds of hectares of marshes and ponds that were present between 1853 and 1901 but no longer exist. Possible causes of those losses include ditching and draining of marshes, wood removal and channel incision, and construction of levees to prevent flooding. While many areas mapped as historical floodplain habitats are now agricultural or developed land, many still flood or accumulate standing water during fall and winter rains. Hence, we conclude that the historical floodplain habitat extent provides a reasonable indication of locations and magnitudes of floodplain restoration opportunities. However, the mapped historical habitat types and locations are not necessarily used as target conditions for restoration. Rather, site-specific target conditions can be guided by the historical potential, but should also consider local restoration goals and constraints imposed by infrastructure or land uses [18, 24].

Contemporary reference information was useful for assessing changes in riparian conditions, as well as changes in habitat conditions resulting from changes in wood abundance, main-channel length, and side-channel length. Contemporary reference sites allowed us to estimate natural potentials that varied among geomorphic settings (floodplain setting or channel type), providing at least a moderate degree of site-specific confidence. In each case we were able to quantify natural potential based on data collected in prior studies [13, 34, 47, 49, 63] or during this study (e.g., wood abundance observations from aerial imagery of the Queets River). Uncertainties in the reference data include unknown effects of the limited land use at or near reference sites, as well as natural variation among reference sites. Use of the contemporary reference information provides somewhat less site-specificity than the historical floodplain habitat mapping, but in all cases we were able to stratify natural potential condition by channel type or channel confinement to reduce extrapolation errors. Location-specific estimates of change in the side-channel and main-channel length have the highest certainty among these habitat features because we made reach-specific estimates of historical length, and compared them to measured current conditions. However, we were still estimating natural

potential condition rather than measuring it as we did with floodplain habitats. Therefore, there is less certainty in the reach-specific values for those changes. For riparian changes we have high certainty in the current condition, which is measured in each reach from lidar or aerial imagery, but moderate certainty in the natural potential estimate, which is estimated for only two riparian strata—non-floodplain and floodplain channels. For habitat changes due to wood loss, both current and natural potential estimates in each reach are mean values for each geomorphic stratum, so reach-specific change estimates have low to moderate certainty at the reach level, but moderate or higher certainty when aggregated to the subbasin level.

For the remaining natural potential estimates we relied on models (fine sediment, temperature, and beaver ponds). The fine sediment and beaver pond models generally have higher uncertainty than the other approaches. For example, the fine sediment model only includes road density as a potential land use driver, and other known sources of fine sediment are not included (e.g., bank erosion or surface erosion from agricultural lands). Hence, the model can give some indication of changes in fine sediment in spawning gravels, but should not be considered a reliable indicator of exactly where fine sediment levels are high or which sources of fine sediment need to be addressed through restoration. The beaver pond model also has poor spatial resolution at the reach level because the estimated natural potential beaver pond density is the same in all small stream reaches [39]. However, models such as these produce more reliable estimates at the subbasin scale because the average pond area across many reaches is more accurate than the estimate for any individual reach. By contrast, we have relatively high confidence in the relative differences in temperature change among reaches because the canopy angle data are based on reach-specific measurements compared to a reference tree height. These reach-specific data then drive the temperature model, so that modeled temperature changes reflect reach-specific differences in canopy opening angle change from natural potential to current conditions.

## Habitat loss and restoration opportunities

The results of the spatial analysis component of the HARP model show that some habitats have been significantly altered from their natural potential conditions (e.g., floodplain marsh and pond habitat, beaver ponds), whereas other habitats are much less altered (e.g., shade or stream channel length). The largest losses are generally in slow water habitat types that provide overwinter habitat and flood refugia for juvenile salmonids, as well as important habitats for other aquatic species. These losses are mostly related to early beaver trapping and draining of marshes for agriculture, and comprise the largest overall habitat change in the Chehalis basin (~90% of those habitats were lost or degraded).

Loss of wood is also assumed to have been substantial given current low abundance in most streams and generally high abundance historically [64, 65], but its effect on habitat change is moderate. For example, decreased spawning gravel areas due to wood loss average roughly 50% across the subbasins, compared to ~90% decrease for the slow water and floodplain habitats. This is because loss of wood does not completely eliminate pools or spawning gravel accumulations, as both features are formed by other processes in the absence of wood [50]. Our riparian assessment showed that most riparian forests in the basin are currently relatively young, suggesting that significant increases in natural wood abundance are not expected until late-century because recovery of wood recruitment does not begin until forests are 60–100 years old [66, 67]. Hence, wood placement is recommended as an interim restoration solution [10], although riparian protection and restoration are important for assuring wood recruitment in the future.

The smallest changes are due to migration barriers and loss of shade. About 22% of all salmon spawning habitat is above any type of artificial barrier, and many of those barriers are modeled with a passage percentage of 33% or 66%, meaning that some or most adult salmon are able to migrate past them. Moreover, some species such as spring- and fall-run Chinook salmon have little or no spawning habitat above barriers in the Chehalis basin because they spawn in larger rivers and streams, and their range is below most or all migration barriers [25]. For loss of shade and resulting temperature changes, only 16% of stream length has a modeled temperature increase >2°C from its natural potential condition, and much of the stream length has a modeled temperature increase <0.5°C. Temperature changes of <0.5°C are likely to have little effect on most juvenile salmon because the temperature differences between optimal and lethal temperatures for juvenile rearing range between 5°C and 9°C [68], and temperature criteria generally do not account for potential use of thermal refugia. However, spawning and rearing ranges and timing for some species are concentrated in areas with current temperatures that are substantially higher than natural potential (e.g., spring-run Chinook salmon). Hence, temperature change may be an important constraint on rebuilding populations of species with high exposure to increased temperatures.

These results suggest that the largest restoration opportunities may be for floodplain and beaver pond habitats, although they are not necessarily important habitats for all species. Among the salmon species, these types of slow-water habitats are particularly important for coho salmon, which use those habitat types extensively for winter rearing [39, 69, 70]. By contrast, steelhead juveniles make little use of those habitat types, and floodplain habitat restoration may provide little benefit. At the other end of the spectrum, shade restoration can reduce temperature between 2°C and 6°C in a limited number of reaches, and barrier removals will provide smaller increases in accessible habitat length. While neither habitat restoration opportunity is widespread, restoration of shade or access can have significant local benefits for species most affected by those issues (e.g., shade-temperature effects on spring-run Chinook salmon or migration barrier effects on coho salmon in specific reaches or subbasins [25]).

In targeting restoration actions based on this assessment, it is important to recognize that the restoration potential is not necessarily the restoration target. Restoration targets and priorities may be less than the full natural potential and may consider other factors such as land use constraints, feasibility, benefit to priority species, and cost of restoration [1, 71]. It is also critical to understand *why* habitats are in poor condition in order to understand *how* to restore them [4, 5]. For example, floodplain reconnection actions differ dramatically when the cause of disconnection is reduction of flow and sediment supply from an upstream dam, versus disconnection by construction of levees at the channel margin. In the former case, restoration may not be possible as long as the dam limits sediment supply, so restoration may focus on sediment augmentation or on dam removal [72, 73]. In the latter case, levee removal or set back are reasonable restoration options [74].

Climate change effects will likely reduce restoration benefits over time, as future climate in the region is expected to produce higher peak flows, lower low flows, and increased stream temperatures [75, 76]. Therefore, restoration plans should consider actions that may ameliorate climate change effects, either by reducing a climate change effect directly or by increasing habitat diversity so fish have more options to avoid the impact [77]. For example, restoring floodplain connectivity can decrease peak flows by increasing water storage [78–80] and reduce stream temperature [38, 81]. Moreover, increasing floodplain connectivity creates flood refugia and thermal diversity [82, 83], which can increase resilience of salmon to climate change by offering more opportunities for fish of all species and life stages to survive extreme events [77]. Similarly, other restoration actions that ameliorate climate change effects such as riparian restoration, summer flow restoration (e.g., reducing irrigation abstraction), re-

aggrading incised channels, and restoring connectivity to diverse habitats can increase resilience of salmon populations to climate change [77, 84]. Including these types of restoration actions in current restoration plans should increase the chance of positive restoration benefits persisting into the future.

## Conclusions

We used the process-based HARP approach to assess habitat restoration potential for each of eight alternative restoration actions in the Chehalis River basin. We used three general methods for quantifying reference conditions for habitats and habitat-forming processes (historical reference, contemporary reference, and models), and diagnosing the degree of alteration for multiple habitat attributes. Each reference condition method provided a reasonable estimate of natural potential condition for analysis, allowing us to estimate declines in both habitat quantity and quality for multiple salmon species. We then use these habitat change results in the habitat analysis and life-cycle model components of HARP [25] to estimate changes in habitat capacity and productivity, and to evaluate how each habitat change has affected salmon populations in the Chehalis River basin. By integrating detailed habitat change analysis with life-cycle models, the HARP approach illustrates how each type of habitat change creates a different restoration potential for each target species, reflecting each species' life history and habitat needs.

## Supporting information

**S1 Fig. Habitat distributions by species.** Distribution of spawning and rearing habitat for each of the four salmon runs in the Chehalis River basin (http://geo.wa.gov/datasets/4ed1382bad264555b018cc8c934f1c01_0). Distributions represent potential distribution, including habitat above man-made migration barriers.
(PDF)

**S2 Fig. Edge unit widths.** Relationship of bar edge, natural bank edge (N bank edge) and armored bank edge (M bank edge) unit widths to total wetted width at 119 transects in the Chehalis River basin. To extrapolate edge unit widths across the Chehalis River basin, we used the regression equations to estimate edge unit widths for all edge habitats based on the mean estimated wetted width of each reach. For bar units and bank units without corresponding hydromodified banks, the historical unit area was considered the same as the current unit area. For modified bank units, area of modified banks represented the current bank habitat, and area of corresponding natural bank (the copy) represented the historical bank habitat.
(PDF)

**S3 Fig. Lidar tree heights by size class.** Box and whiskers plots of lidar tree heights in each size class identified on aerial photography (bar is the median, box represents the 25th to 75th percentiles, whiskers represent the 10th and 90th percentiles, and circles are "outliers"). Each point represents one point at which an observer classified a tree size class on aerial photography and tree height in the lidar data set. There were two observers, and for each observer there were a total of 232 point measurements at 116 sites (one point on each side of the stream at each site), so there were a total of 464 points. Sample sizes for each size class are: Tall = 29, Medium = 307, Short = 104, and No veg = 24.
(PDF)

**S4 Fig. Geomorphic settings for riparian reference conditions.** For riparian reference conditions (e.g., the natural potential tree height), we stratified the basin into non-floodplain channels with stable riparian landforms (terraces or hill slopes, upper panel), and floodplain

channels with varying rates of lateral channel migration and floodplain turnover. Channels with narrow or no floodplain are typically dominated by upland forest types in western Washington [1, 2]. Floodplain channels have floodplains >4 times the width of the main channel, and multiple side channels may flow across the floodplain [3, 4]. Because these channels constantly erode floodplain surfaces at one location and create new ones at other locations, the riparian forest consists of many small stands of varying ages and species compositions [5]. (PDF)

**S5 Fig. Reference conditions for non-floodplain and floodplain channels.** Non-floodplain channels have longer disturbance return intervals relative to floodplain channels, and conifer-dominated forests. The left panels show cumulative stand age distributions under modeled fire return intervals and forest management for non-floodplain channels [1] and erosion return intervals for floodplain channels [2]. Right panels show species compositions by land form for floodplain and non-floodplain channels [3, 4]. Natural potential tree heights used in the model were 52 m for non-floodplain channels and 30.5 m for floodplain channels based on tree heights for Douglas-fir for non-floodplain channels and red alder for floodplain channels [5]. (PDF)

**S6 Fig. Canopy opening angle diagram.** Illustration of canopy opening angle ($\theta$) and the parameters used to calculate it [1]. Left bank tree height and right bank tree height are $z_L$ and $z_R$, respectively, and $W$ is bankfull channel width. The equation for calculating the canopy opening angle is: $\theta = \left(90 - tan^{-1}\left(\frac{z_L}{\frac{W}{2}}\right)\right) + \left(90 - tan^{-1}\left(\frac{z_R}{\frac{W}{2}}\right)\right)$ The inverse tangent functions are subtracted from 90˚, so a channel with complete canopy closure will have $\theta = 0$˚ and a channel with no vegetation on either bank will have $\theta = 180$˚. (PDF)

**S7 Fig. Fine sediment analysis.** (A) Comparison of fine sediment values and road density from the Chehalis (circles) [1] to published values of fine sediment plotted against road density for the Queets and Clearwater Rivers on the Olympic Peninsula (triangles). The line represents the linear model of percent fines to road density [2]. Note that samples from Scatter Creek and Mima Creek tributary (Mima trib) have particularly high percent fines at relatively low road density. (B) Percent fines plotted against the shear stress index (*slope × bankfull width*). Below the threshold of 0.05 (dotted line), the average fine sediment of the six sites is 27.6% fines. We estimated road density using the Flow Accumulation tool, which counts all cells upstream of any given cell, as well as the number of road cells upstream of that cell. An accuracy test showed that this more efficient method produced road area consistently 1.3 times higher than a more accurate but computationally intensive line-derived road area. Therefore, we used a correction factor of 0.767 on all raster road areas to increase accuracy. (PDF)

**S8 Fig. Map of historical floodplain habitats including marshes, ponds, and lakes.** (PDF)

**S1 Table. Large river main channel and side channel length multipliers.** Table shows all values for the mainstem Chehalis and Chehalis tributaries by river reach. We developed historical length multipliers for each reach based on unpublished lidar-derived data from Natural Systems Design (NSD) and reference values of sinuosity and side-channel length [1, 2]. (PDF)

**S2 Table. Mean percent pool area by slope class and land cover class.** The 'Reference' column shows mean percent pool for the historical period, whereas land cover classes show

current mean percent pool for each land cover and slope class.
(PDF)

**S3 Table. Beaver pond densities.** Estimated pond densities from maps and data showing calculation of weighted mean natural potential beaver ponds per km in the Chehalis basin [1]. Number of ponds is the manually-counted number of dam symbols in each subbasin from the published maps.
(PDF)

## Acknowledgments

We greatly appreciate the numerous people who help collect and analyze data for this project including Jamie Thompson, Gus Seixas, Josh Chamberlin, Jason Hall, Peter Kiffney, Spencer Kubo, and Jenna Keaton, as well as collaboration and review from Larry Lestelle, Gary Morishima, Neala Kendall, John Ferguson, Chip McConnaha, Jon Walker, Laura McMullen, Mara Zimmerman, John Winkowski, Tim Abbe, and Susan Dickerson-Lange. Their help along the way was critical to the completion of this project. Helpful reviews of the manuscript were provided by Jeff Jorgensen, Aimee Fullerton, Morgan Bond, and one anonymous reviewer.

## Author Contributions

**Conceptualization:** Timothy J. Beechie.

**Data curation:** Caleb Fogel, Colin Nicol, Britta Timpane-Padgham.

**Formal analysis:** Caleb Fogel, Colin Nicol, Britta Timpane-Padgham.

**Funding acquisition:** Timothy J. Beechie.

**Methodology:** Timothy J. Beechie.

**Project administration:** Timothy J. Beechie.

**Supervision:** Timothy J. Beechie.

**Writing – original draft:** Timothy J. Beechie, Caleb Fogel.

**Writing – review & editing:** Timothy J. Beechie, Caleb Fogel, Colin Nicol, Britta Timpane-Padgham.

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
