## [Decision Letter · Decision Letter 0]

26 May 2021

PONE-D-20-37830

A process-based assessment of landscape change and salmon habitat losses in the Chehalis River basin, USA

PLOS ONE

Dear Dr. Beechie,

Thank you for submitting your manuscript to PLOS ONE. After careful consideration, we feel that it has merit but does not fully meet PLOS ONE’s publication criteria as it currently stands. Therefore, we invite you to submit a revised version of the manuscript that addresses the points raised during the review process.

The authors merely need to address the few questions posed in the detailed review and make necessary changes. These comments are minor and should not take a great deal of time.  Overall this manuscript is well written and executed.

We look forward to receiving your revised manuscript.

Kind regards,

Madison Powell, PhD

Academic Editor

PLOS ONE

Additional Editor Comments (if provided):

The authors of this manuscript provide worthwhile data and provide a detailed summation of a variety of data regarding habitat types etc. The manuscript is well written and generally easy to follow. The manuscript's information is of interest to a variety of fish biologists and habitat biologists outside of salmonid biology. The authors need to address the comments provided in the detailed review and answer the questions posed by the reviewer.

.

Journal Requirements:

3) Our internal editors have looked over your manuscript and determined that it is within the scope of our Freshwater Ecosystems Call for Papers. This collection of papers is headed by a team of Guest Editors for PLOS ONE (https://collections.plos.org/s/freshwater-ecosystems). The Collection will encompass a diverse range of research articles on freshwater ecology, including freshwater ecosystems conservation. Additional information can be found on our announcement page: https://collections.plos.org/s/freshwater-ecosystems.

If you would like your manuscript to be considered for this collection, please let us know in your cover letter and we will ensure that your paper is treated as if you were responding to this call. If you would prefer to remove your manuscript from collection consideration, please specify this in the cover letter.

4) Thank you for stating the following in the Competing Interests section:

[The authors have declared that no competing interests exist.].   

We note that one or more of the authors are employed by a commercial company: Ocean Associates, Inc and A.I.S., Inc.

i. Please provide an amended Funding Statement declaring this commercial affiliation, as well as a statement regarding the Role of Funders in your study. If the funding organization did not play a role in the study design, data collection and analysis, decision to publish, or preparation of the manuscript and only provided financial support in the form of authors' salaries and/or research materials, please review your statements relating to the author contributions, and ensure you have specifically and accurately indicated the role(s) that these authors had in your study. You can update author roles in the Author Contributions section of the online submission form.

ii. Please also provide an updated Competing Interests Statement declaring this commercial affiliation along with any other relevant declarations relating to employment, consultancy, patents, products in development, or marketed products, etc.  

5) We note that you have stated that you will provide repository information for your data at acceptance. Should your manuscript be accepted for publication, we will hold it until you provide the relevant accession numbers or DOIs necessary to access your data. If you wish to make changes to your Data Availability statement, please describe these changes in your cover letter and we will update your Data Availability statement to reflect the information you provide.

6) Please include captions for your Supporting Information files at the end of your manuscript, and update any in-text citations to match accordingly. Please see our Supporting Information guidelines for more information: http://journals.plos.org/plosone/s/supporting-information.

7) We note that Figures in your submission contain map/satellite images which may be copyrighted. All PLOS content is published under the Creative Commons Attribution License (CC BY 4.0), which means that the manuscript, images, and Supporting Information files will be freely available online, and any third party is permitted to access, download, copy, distribute, and use these materials in any way, even commercially, with proper attribution. For these reasons, we cannot publish previously copyrighted maps or satellite images created using proprietary data, such as Google software (Google Maps, Street View, and Earth). For more information, see our copyright guidelines: http://journals.plos.org/plosone/s/licenses-and-copyright.

i.    You may seek permission from the original copyright holder of Figure(s) [#] to publish the content specifically under the CC BY 4.0 license. 

ii.    If you are unable to obtain permission from the original copyright holder to publish these figures under the CC BY 4.0 license or if the copyright holder’s requirements are incompatible with the CC BY 4.0 license, please either i) remove the figure or ii) supply a replacement figure that complies with the CC BY 4.0 license. Please check copyright information on all replacement figures and update the figure caption with source information. If applicable, please specify in the figure caption text when a figure is similar but not identical to the original image and is therefore for illustrative purposes only.

Reviewers' comments:

Reviewer's Responses to Questions

**Comments to the Author**

1. Is the manuscript technically sound, and do the data support the conclusions?

Reviewer #1: Yes

2. Has the statistical analysis been performed appropriately and rigorously? 

Reviewer #1: Yes

3. Have the authors made all data underlying the findings in their manuscript fully available?

Reviewer #1: Yes

4. Is the manuscript presented in an intelligible fashion and written in standard English?

Reviewer #1: Yes

5. Review Comments to the Author

Reviewer #1: General

This manuscript is part of a multi-paper contribution outlining a framework to estimate 1. historical habitat conditions compared to present day and 2. which changes are most likely to limit fish recovery. This manuscript addresses goal #1 of this framework by quantifying changes in the amount of different habitat types. The authors analyze an impressive range of data to characterize historical conditions, and they find that marsh habitats have seen the largest reductions compared to baseline conditions. While identifying limiting factors via life-cycle modeling is a potentially contentious exercise that can be difficult to parameterize with existing data, characterizing changes in habitat quantity is generally less complex and contentious (though it is important and certainly not easy to pull off). I found this paper was easy enough to read and it clearly supported its claims with robust analyses.

Specific

77: authors do a nice job placing the objectives of this paper in the context of the 3-step approach

85: If the restoration is process-based, why is the measure of impact pattern based?

88: How about natural conditions varying temporally, given giant variation in salmon production prior to contemporary land use (e.g., Rogers et al. PNAS 2013 Centennial-scale fluctuations)? Could help to briefly address temporal variation when introducing spatial variation, so it doesn’t come off as assuming that the past was static.

90: could end with ‘,respectively’ I had to read this twice to understand this sentence.

99: Does ‘dial’ imply that they can be manipulated by management?

134: Is there typically enough data to quantify population bottlenecks in terms of the habitat features being restored? For example, if marshes provided flood refuge or spring foraging, when is there the data to understand how area of marsh and fish density translate into survival? Isn’t the resolution of data more coarse than that needed to address the processes and functions being restored? [OK I see that the life-cycle paper is separate from this one, so this comment doesn’t need to be addressed in this manuscript]

221: Many readers may wonder how climate change fits in with this and how we assess baseline conditions if they unlikely to occur in future climates.

Fig. 1 Why is floodplain connectivity a driver? Isn’t this a habitat condition? The Driver/Habitat conditions differentiation isn’t that clear for me, so it could be explained a little more when introduced…. Or the terms could be reworded to sound like drivers. For example floodplain connectivity (a habitat condition in my mind) could be rephrased so that it was parallel with terms such as road density and channel straightening. But in general the mix up between mechanisms and patterns here is confusing.

6. PLOS authors have the option to publish the peer review history of their article (what does this mean?). If published, this will include your full peer review and any attached files.

Reviewer #1: No

---

## [Author Response · Author response to Decision Letter 0]

20 Jul 2021

All comment responses are included in the Response to Reviewers and the Cover Letter

---

## [Editor Report · Decision Letter 1]

23 Sep 2021

A process-based assessment of landscape change and salmon habitat losses in the Chehalis River basin, USA

PONE-D-20-37830R1

Dear Dr. Beechie,

We’re pleased to inform you that your manuscript has been judged scientifically suitable for publication and will be formally accepted for publication once it meets all outstanding technical requirements.

Kind regards,

Madison Powell, PhD

Academic Editor

PLOS ONE

Additional Editor Comments (optional):

Minor revisions are complete. Authors have addressed all editorial comments.
---

## [Editor Report · Acceptance letter]

21 Oct 2021

PONE-D-20-37830R1 

A process-based assessment of landscape change and salmon habitat losses in the Chehalis River basin, USA 

Dear Dr. Beechie:

I'm pleased to inform you that your manuscript has been deemed suitable for publication in PLOS ONE. Congratulations! Your manuscript is now with our production department. 

Kind regards, 

on behalf of

Dr. Madison Powell 

Academic Editor

PLOS ONE